# Application of Machine Learning Methods to Investigate Joint Load in Agility on the Football Field: Creating the Model, Part I

**DOI:** 10.3390/s24113652

**Published:** 2024-06-05

**Authors:** Anne Benjaminse, Eline M. Nijmeijer, Alli Gokeler, Stefano Di Paolo

**Affiliations:** 1Department of Human Movement Sciences, University Medical Center Groningen, University of Groningen, 9713 AV Groningen, The Netherlands; e.m.nijmeijer@umcg.nl; 2Exercise Science and Neuroscience Unit, Department of Exercise & Health, Faculty of Science, Paderborn University, 33098 Paderborn, Germany; alli.gokeler@uni-paderborn.de; 3Orthopedic and Traumatologic Clinic II, IRCCS, Istituto Ortopedico Rizzoli, 40136 Bologna, Italy; stefano.dipaolo@ior.it

**Keywords:** anterior cruciate ligament, kinetics, on field, IMU, injury prevention, female football, ecological dynamics approach, athlete–environment relationship

## Abstract

Laboratory studies have limitations in screening for anterior cruciate ligament (ACL) injury risk due to their lack of ecological validity. Machine learning (ML) methods coupled with wearable sensors are state-of-art approaches for joint load estimation outside the laboratory in athletic tasks. The aim of this study was to investigate ML approaches in predicting knee joint loading during sport-specific agility tasks. We explored the possibility of predicting high and low knee abduction moments (KAMs) from kinematic data collected in a laboratory setting through wearable sensors and of predicting the actual KAM from kinematics. Xsens MVN Analyze and Vicon motion analysis, together with Bertec force plates, were used. Talented female football (soccer) players (n = 32, age 14.8 ± 1.0 y, height 167.9 ± 5.1 cm, mass 57.5 ± 8.0 kg) performed unanticipated sidestep cutting movements (number of trials analyzed = 1105). According to the findings of this technical note, classification models that aim to identify the players exhibiting high or low KAM are preferable to the ones that aim to predict the actual peak KAM magnitude. The possibility of classifying high versus low KAMs during agility with good approximation (AUC 0.81–0.85) represents a step towards testing in an ecologically valid environment.

## 1. Introduction

An injury to the anterior cruciate ligament (ACL) is severe and requires surgical intervention in many cases [1]. Per year, over two million ACL injuries occur worldwide. The majority of these injuries are observed in pediatric and adolescent athletes [1,2,3,4,5], with a higher incidence in girls compared to boys, especially at younger ages [1,6,7]. These ACL injuries lead to the longest withdrawal time from youth sports. Only 44% of these young athletes return to their pre-injury level of sports, and up to 23% sustain a re-injury, with many of them dropping out of sports entirely [8]. An ACL rupture is a devastating injury for a soccer player, resulting in the longest layoff times [9] and a reduced career length [10]. This change in life immensely affects both the athletes’ physical and psychosocial well-being in the short and long term. Many of the health benefits of playing sports are lost due to injuries, negatively affecting a healthy lifestyle and public health.

Most ACL injuries are non-contact injuries caused while approaching an opponent using a single leg deceleration during a sidestep cutting (SSC) maneuver [11]. Video analyses have shown frequent hip abduction, knee abduction and the external rotation of the knee combined with a flat foot or heel strike pattern while rupturing the ACL [11]. While laboratory studies have been used to identify the modifiable biomechanical and neuromuscular risk factors associated with ACL injuries in female football players during SSC over the last two decades, they also have limitations due to their lack of ecological validity. These studies conducted in controlled settings offer valuable insights into motor control and biomechanics. Nonetheless, they fall short in capturing the full complexity and variability of real-world movements [12]. Factors such as the terrain, weather conditions, and environmental stimuli like the presence of other players and the ball are typically not replicated in laboratory settings, thereby restricting the ecological validity of the research findings [12]. In the real world, players have to anticipate and respond to unpredictable external stimuli, inhibit impulsive actions [13], and coordinate quickly and precisely to achieve optimal movement patterns during changes of direction (CODs) [14].

Thus, to establish the true etiology and mechanism of ACL injuries, one needs to consider the real-world environment [12]. For example, using inertial measurements (IMUs) on the football field allows for more ecologically valid measurements. This has recently shown that female football players indeed adopt a context-related motor strategy, indicating kinematical COD differences between the laboratory and the football field [15]. Also, video analyses have provided us with kinematic patterns of the context-specific ACL injury situations that occur in female football during defending COD scenarios [11]. However, this research has so far only given us insight into joint kinematics, without considering the cause of the movement. On-field joint kinetics would allow for a more informed understanding of the movement patterns in ecologically valid situations, as it describes the forces that cause the movement and could potentially increase the risk of ACL injury [16,17]. Machine learning (ML) approaches could assist in estimating joint loads directly using real-world data [15]. This would provide a further step towards obtaining an athlete’s comprehensive biomechanical profile in a sport-specific context [18]. For this, we determined that the joint angle waveforms of COD tasks captured on the field could be trustfully replicated within the laboratory benchmark [15]. This was crucial to set up robust ML algorithms replicating the joint loads computed in the laboratory (using a force plate) using data captured on the field (without a force plate) [11]. Regression models and ML are state-of-art statistical approaches for joint load estimation outside the laboratory [19]. So far, there is lack of agreement in the literature on which is the best classification or regression model to derive kinetics metrics from IMU kinematics, and the inspection of multiple models is a common practice [20,21].

The aim of this study is therefore to investigate ML approaches, to be able to predict knee joint loading in the near future during on-field COD tasks, i.e., agility. To do so, we explore the possibility of (I) predicting high and low knee abduction moments (KAMs) from kinematics data collected in a laboratory setting through wearable sensors, and (II) predict the actual KAM from kinematics. Several machine learning models are compared in order to explore the best solution to either the classification or regression problems. The models produced here can then be used on the field to predict knee joint loading in an ecological environment. This can then be interpreted as a benchmark for the female footballers, with a protective motor strategy for the ACL injury, and used as an example for more at-risk players. By leveraging technology to assess the biomechanics of female football players on the field, this study has the potential to provide comprehensive, context-specific insights into the at-risk movement behaviors of female players. The findings can contribute to the advancement of injury prevention in female football.

## 2. Materials and Methods

### 2.1. Participants

All procedures were approved by the Medical Ethical Committee of the Blinded for submission (ID number: Blinded for submission). All players and their parents/legal guardians signed an informed consent form before inclusion. Talented female football (soccer) players (n = 32, age 14.8 ± 1.0 y, height 167.9 ± 5.1 cm, mass 57.5 ± 8.0 kg) were included. All players were part of a regional talent training program. Players’ engagement consisted of four to five training sessions (average training session time: 75 min) and one official game per week. Players’ dominant leg was identified as the preferred leg to jump and land with [15], with 28 players identified as right dominant.

### 2.2. Data Collection

Data collection was performed during regular football seasons (September–May). Two measurement systems were used to collect the data: (1) the Xsens MVN Analyze system (Xsens Technologies, Enschede, The Netherlands) and (2) the Vicon motion system (Vicon Motion Systems, INC., Centennial, CO, USA), together with Bertec force plates (Bertec Corporation Columbus, OH, USA). The participants wore an MVN Lycra suit consisting of 17 inertial measurement units (IMUs), a battery, and a transmission pack inside the compression suit. The IMUs were placed on the right side of the head, the sternum, the posterior side of the hands, the dorsal side of the wrists, the lateral side above the elbows, the middle of the scapula spines, the middle of both the posterior superior iliac spines (pelvis), the lateral side of the thighs, the medial surface of the tibias and the dorsal side of the forefoot. Each IMU was integrated with a 3D magnetometer that internally sampled at 1 KHz, with an overall sample rate of 240 Hz [22]. After putting on the suit, the anthropometric measures of Xsens and Vicon were taken. Sixteen reflective markers were placed according to the Vicon Plug-in-Gait lower body model (Vicon Motion Systems, INC., Centennial, CO, USA). Five additional trunk markers were placed on the sternum, clavicle, C7, T10, and right scapula. N-pose + walk (Xsens) and T-pose (Vicon) calibration were performed according to the manufacturer’s recommendation.

### 2.3. Agility Task

The players executed unanticipated sidestep cutting movements as described earlier [15,23]. In short, the players used a 5 m approach run followed by a 1 foot landing with the preferred leg, and a 40–50° change in direction; this was followed by running through a gate 5 m away. The task was a mirror exercise in which the player had to respond as quickly and accurately as possible to unanticipated changes of direction by a buddy. All players followed an intervention program consisting of a baseline test, four training sessions (separated by one week), an immediate posttest after the fourth training session and a one-week retention test [24]. During the tests, five trials of the task were collected. Ten trials were collected during each training session, resulting in a total of 55 trials for each player.

### 2.4. Data Preparation and Feature Extraction

Kinetic data concerning the knee in all three directions were extracted and filtered with a fourth-order zero-lag Butterworth low-pass filter at 10 Hz [23]. The external knee moments were normalized to body mass. The time window was normalized from 0% to 100% according to the contact of the force platform (5% of the peak force values were used as cut off for the initial and final contact frame). Since the highest knee moments are usually registered during the first impact on the ground, the peak knee abduction moment (KAM) was extracted for each trial, as well as the normalized frame of its occurrence within the 0–20% of the cut foot stance [25]. The 33rd and 67th percentiles of the peak KAM were computed as the boundaries of “low” and “high” KAMs, respectively.

The knee, hip, ankle, and pelvis joint kinematics in all three planes was extracted from the Xsens MVN Analyze 2020.0.1 software suit (Xsens Technologies, Enschede, Netherlands) and processed using a customized script in Matlab (The MathWorks, Natick, MA, USA). The joint angles were defined using the Euler sequence ZXY. The following features were extracted from each joint angle in each anatomical plane: (I) mean angle over 0–20% of the cut foot stance; (II) peak angle over 0–20% of the cut foot stance; and (III) angle at the frame of the peak KAM. These three distinct kinematics metrics were chosen in order to explore the most feasible kinematics series of features to be related to the KAM. In particular, mean kinematics (0–20%) was chosen because it nicely incorporates the foot stance phase with the maximum knee loading after initial contact with the ground and the phase at which the ACL injury is thought to happen according to previous video analysis studies [25]. The peak kinematics feature was chosen because of its ease of collection and its potential to identify kinematics risk factors. Lastly, kinematics at the peak KAM was adopted to identify the kinematics at the exact time frame of the supposed peak knee load; this last kinematics feature is, if confirmed to be a valuable tool to classify high and low KAM, the closest kinematics dataset to the definition of high and low KAM. The adoption of these three features thus allows a broader understanding of the lower-limb kinematics during the cutting maneuver according to KAM production. So far, no clear superiority of one of these features as an input measure to inform ML models during ACL injury risk prediction has been identified [26].

### 2.5. Machine Learning Method: Classification Analysis

A classification model for each of the three kinematics features (mean kinematics, peak kinematics, kinematics at peak KAM) was developed in Matlab (Classification Learner app, vR2022a, The MathWorks, Natick, MA, USA). The peak KAM value was dichotomized in high (+1) or low (−1) and was used as the dependent variable. In each model, 12 independent predictors (four joints, three anatomical planes) were included. A validation step and a test step were conducted consequently. The validation step was conducted on 80% of the total dataset and the remaining (randomly selected) 20% was used in the test step [27]. A dimensionality reduction step was conducted before the validation step, and this used principal component analysis (PCA). The minimum number of model predictors required to explain 95% of the total variance was kept [28]. A 5-fold cross-validation step was adopted to protect against overfitting while maximizing data utilization, in accordance with previous literature with similar rationale and best practices [29,30]. The accuracy outcomes obtained from the cross-validation were reported as the average across all folds. A total of 32 classification models were tested: six support vector machines (SVMs), six nearest neighbor classifiers, five ensemble classifiers, five neural network classifiers, tree decision trees, two discriminant analysis, two Naïve Bayes classifiers, two kernel approximation classifiers, and one logistic regression classifier. The full list of models is presented in Appendix A. The classification model with the highest validation accuracy and area under curve (AUC) in the receiving operating curve (ROC) analysis was used in the model testing. The following outcomes were considered to inspect the model performance in the test phase: accuracy, AUC, true positive rate (TPR), and positive predictive values (PPVs).

### 2.6. Machine Learning Method: Regression Analysis

A regression model for each of the three kinematic features (mean kinematics, peak kinematics, kinematics at peak KAM) was developed in Matlab (Regression Learner app). The actual peak KAM value was used as a dependent variable. In each model, 12 independent predictors (4 joints, 3 anatomical planes) were included. A validation step and a test step were conducted consequently. A dimensionality reduction step was conducted before the validation step, which adopted principal component analysis (PCA). The minimum number of features required to explain 95% of the total variance was kept [28]. A 5-folds cross-validation step was adopted also in this analysis and accuracy average across all folds was presented. 

The validation step was conducted on 80% of the total dataset and the remaining (randomly selected) 20% was used in the test step [27]. A total of 26 regression models were tested: six SVM, five neural networks, four Gaussian Process Regression (GPR) models, four linear regression models, three regression trees, two ensembles of trees, and two kernel approximation models. The full list of models is presented in Appendix A. The regression model with the lowest validation root mean square error (RMSE) and the highest R2 was used in model testing. The following outcomes were considered to inspect the model performance in the test phase: RMSE and R2. The complete flowchart of the data extraction and processing is presented in Figure 1.

### 2.7. Statistical Analysis

Continuous data were presented as the mean and standard deviation, while categorical data were presented as frequency and percentage over the total. The Student’s *t*-test was used to compare the joint kinematics (mean, peak, and value at peak KAM) between high and low KAMs (defined according to high and low tertiles, respectively). Differences were considered statistically significant if *p* < 0.05. Classification models with accuracy, AUC, TPR, and PPV ≥80%, and regression models with R2 ≥ 0.5, were considered acceptable [26,27,31]. All statistical analyses were conducted in Matlab.

## 3. Results

Overall, 1285 trials were included in the analysis. In total, 475 trials were excluded because of (1) missing kinematics and/or kinetics data due to technical issues during acquisition or processing in at least one of the two systems, or (2) failure to complete the training program due to COVID-19 or injuries sustained during other sport activities. The KAM waveforms for high, middle and low KAMs are presented in Figure 2. The average KAM was 1.1 ± 1.2 Nm/BW and occurred at 7.4 ± 4.7% of the cut foot stance. From this dataset, the middle tertile of the peak KAM (peak KAM > 33rd percentile and <67th percentile, n = 375) was removed for the rest of the processing. Therefore, 730 trials (low KAM n = 365, avg = 0.24 Nm/BW; high KAM n = 365, avg = 1.59 Nm/BW) were included in the classification and regression models.

### 3.1. Descriptive Kinematics According to Knee Abduction Moment

In the trials classified as having a high KAM, the players showed greater knee flexion, hip flexion, hip adduction, ankle dorsiflexion, and lower ankle eversion and internal rotation compared to those in the trials classified as having a low KAM in mean kinematics, with between 0–20% of foot stance (Table 1). Similar differences emerged also in the peak kinematics and the kinematics at the peak KAM, with the addition of lower pelvis anteversion and rotation in the former and the latter, respectively (Table 1).

### 3.2. Classification Analysis

After PCA, 8 (mean kinematics and kinematics at peak KAM) to 9 (peak kinematics) predictors out of 12 were kept in the ML analysis. The best model in the validation step was the Fine Gaussian SVM for all three kinematics features. In the test step, the model accuracy ranged from 72.6% (peak kinematics) to 79.5% (mean kinematics). The AUC (Figure 3) ranged from 0.81 (peak kinematics) to 0.85 (mean kinematics), the TPR ranged from 72.6% (value at peak KAM) to 89.0% (mean kinematics), and the PPV ranged from 69.4% (peak kinematics) to 79.1% (value at peak KAM). A detailed description of the classification model performances for the three kinematics features is presented in Table 2.

### 3.3. Regression Analysis

After PCA, 8 predictors out of 12 were kept in the ML analysis for all three kinematics features. The best model in the validation step was either the Matern 5/2 GPR (mean and peak kinematics) or the Exponential GPR (kinematics at peak KAM). In the test step, the RMSE ranged from 0.936 (value at peak KAM) to 1.078 Nm/BW (peak kinematics). The R2 ranged from 0.33 (peak kinematics) to 0.46 (mean kinematics). A detailed description of the regression model performances for the three kinematics features is presented in Table 3.

## 4. Discussion

The most important finding of the present study was the good accuracy of the ML model in classifying agility movement tasks performed with high or low KAMs in laboratory settings. Such a model has the potential to identify high joint loads during the agility testing of football players based on their joint kinematics, collected through wearable sensors. This model could therefore be used to inspect on-field players’ motion and inform ACL injury prevention strategies. A second finding of the study was the suboptimal accuracy of the regression models in predicting the actual KAM peak value associated with agility task kinematics. Based on these findings, caution should be exercised when discerning the peak KAM values from on-field kinematics and clinical recommendations should be provided accordingly.

First, the classification models showed an overall accuracy of about 80% and an AUC of 0.81–0.85 according to the kinematic features adopted (Table 2). Thus, the models could classify the agility tasks performed with high or low KAMs based on players’ kinematics with “acceptable” model performances [26,27,32]. According to a proper input including the full-body joint kinematics patterns obtained from an agility task, the model could therefore highlight the presence of high knee joint loading in a football player and provide precious insights on her higher risk of sustaining an ACL injury. Among the wide set of classification models explored, the SVM Fine Gaussian model showed the best accuracy for all the features. This is in line with the current literature, considering the SVM approach as a proper solution to dealing with high dimensionality problems with a high discriminative classification power in sport applications [33,34]. In terms of kinematic features, the mean kinematics in the first 0-20% of the stance phase showed the best accuracy in model training and testing. Such a window describes the kinematic response to the initial contact loads and roughly has the peak KAM at its center (7.4 ± 4.7% of the cut stance phase). Also, a TPR of about 90% was noted in the model testing, thus confirming its excellent performance in identifying agility trials that generate high KAMs. This feature is therefore a comprehensive compromise between the collection of isolated peak kinematic values and the whole kinematic waveform during the foot stance. It is also easy to collect during on-field testing.

Second, the regression models showed suboptimal accuracy in the prediction of the peak KAM magnitude. Less than 50% of the variance was explained by the best models for each of the kinematic features (R2 0.33–0.46, Table 3). Therefore, none of the models reached an acceptable threshold in the model performance [27,32]. Moreover, an RMSE of 0.94–1.1 Nm/BW was noted (Table 3). Such an error shall be considered high, given that the mean peak KAM emerged from the cohort of this study, even for the high KAM subgroup (1.6 Nm/BW). Notably, as for the classification model, the best results were obtained when the mean kinematics features were used as model predictors. The use of the peak KAM magnitude predicted from agility kinematics is therefore not recommended for on-field use. A possible reason for the suboptimal performances that emerged is the overall magnitude of the KAM, which typically ranges from 0 to 2.5 Nm/BW [35]. Moreover, absolute knee joint moments could also be influenced by other factors such as the speed and height of the players. The identification of actual magnitude peak joint kinetics from joint kinematics is a common bottleneck in the literature regarding sports movements and ML approaches [27].

In this study, we have created a model that is now able to classify high versus low KAM during agility with good approximation based on kinematic data. This represents a step towards testing in an ecologically valid environment. In the past, we have advocated for preserving the task–athlete–environment relationship [12], as movement emerges, as per definition, from interaction with the environment [36]. This allows us to move away from in-lab injury risk screening. This is essential, as the information we receive from such constrained types of testing in sterile laboratory environments is not valid [23]. Previously, we have stated that the adoption of ecological testing will enable football coaching and medical staff to effectively reduce the occurrence of high-risk biomechanical patterns [15]. The model developed in this technical note will move ACL injury screening prevention practices forward. In the future, we will transfer this technology and methodology towards a user-friendly version with, e.g., a minimal sensor set-up, so that it can become a standard approach for research and practical on-field use. Identifying players at a heightened risk of ACL injury will aid staff (i.e., medical staff, coaches, and strength and conditioning specialists) in developing a more personalized and targeted approach to injury prevention training, taking into account the complex interplay of factors that contribute to injury susceptibility in female football players. Additionally, ML could be used to predict the likelihood of an athlete suffering an ACL injury based on their movement patterns and other factors, allowing for proactive measures to be taken to prevent ACL injury. By developing player profiles [23] and classifying players into high versus low KAM during agility, we will work towards a final application, embracing technology that is right around the corner.

To the best of the authors’ knowledge, the present study was the first to adopt an ML model in the assessment of KAMs using full-body joint kinematics in a large dataset (n = 1285) of agility tasks. A randomly selected portion of the original dataset was used to test the performances of the models, PCA was adopted to reduce the data dimensionality, and overfitting issues were minimized by cross-validation. A large number of either classification (n = 32) or regression (n = 26) models were also explored. Moreover, clinically relevant kinematics features that were collected through wearable inertial sensors were adopted in order to maximize the models’ applicability to an on-field scenario. Still, this study has some limitations. First, the peak and mean kinematics and kinetics values were considered as either the input or output of the ML models in this study. No continuous waveform data were included in order to minimize the model complexity and favor the interpretability of the potential on-field use. Future studies could investigate the adoption of waveform data both for kinematics and kinetics as input and output, respectively. Second, data belonging to the mid tertile of the KAMs were excluded from the ML models. This choice reduced the number of trials included in the classification and regression analyses. However, this was performed to facilitate the clinical interpretation of the findings and to promote the practical adoption of the current methods in clinical practice by removing the “gray area” of KAM magnitude trials. Still, our research constitutes a large set of trials compared to others [37,38,39]. Third, the trials were collected in different time frames and there were different numbers per subject, according to the main project study protocol [24]. However, this constitutes a minor limitation, and even enhances the intra- and inter-subject variability of the input data for the ML models. In addition, the results of this study cannot be extrapolated to other ages, levels, sports and males. For example, it should be taken into account that girls at this age mature; therefore, their risk profiles may look different after, e.g., the peak height velocity has been reached. Also, we normalized the kinematics in the cut stance time window from 0% to 100% for every trial. Lastly, screening based on a full-body IMU set is expensive and not practical. Eventually, this line of research will inform practices concerning on-field joint loads. Our goal is then to transform this into usable information for clinicians and coaches.

## 5. Conclusions

According to the findings of this technical note, classification models that aim to identify the players exhibiting high or low KAMs are preferable to the ones that aim to predict the actual peak KAM magnitude. In order to promote the clinical implementation of such models in ACL injury prevention in more ecologically valid scenarios, the occurrence of false positives should be minimized, as well as the absolute error margin. Our study provides a next step in on-field ACL injury risk screening in youth female soccer players. The possibility of classifying high versus low KAMs during agility with good approximation based on kinematic data represents a step towards testing in an ecologically valid environment.

## Figures and Tables

**Figure 1 sensors-24-03652-f001:**
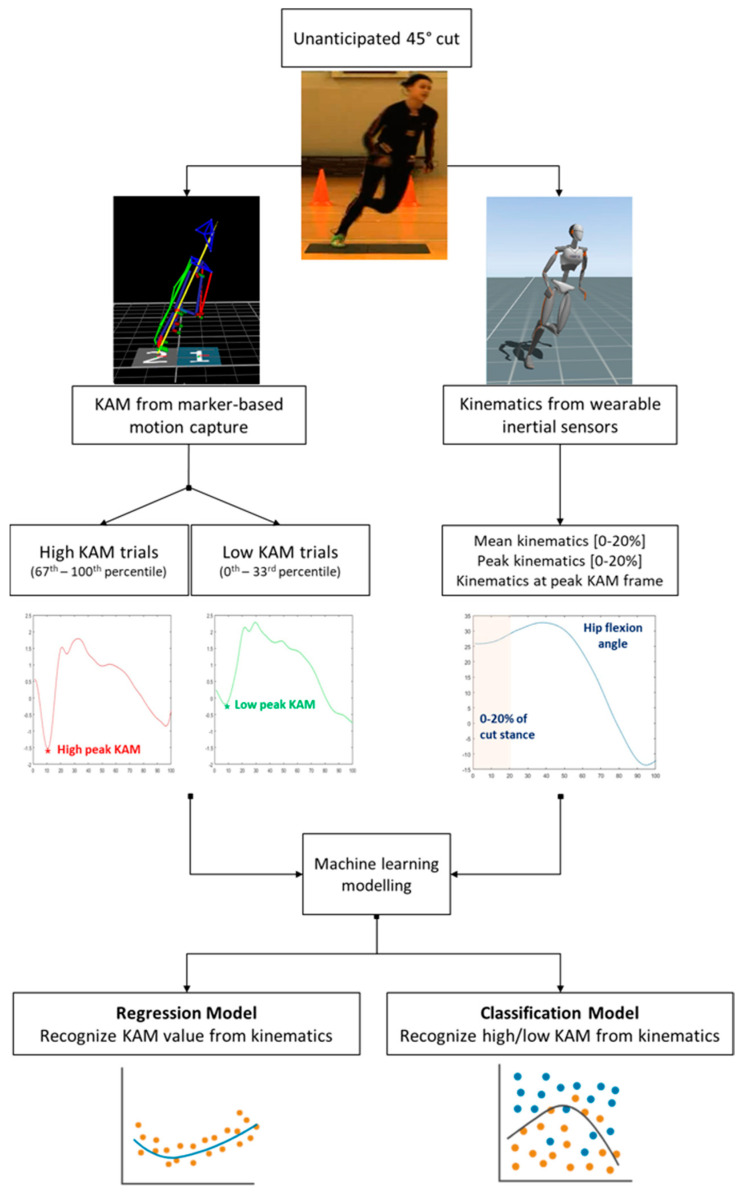
Flowchart of data extraction and processing.

**Figure 2 sensors-24-03652-f002:**
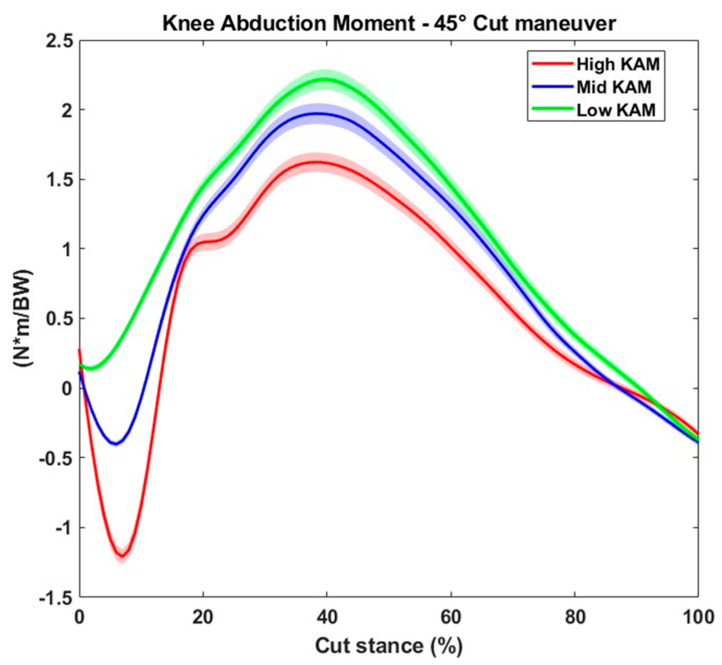
Waveforms for high, middle and low knee abduction moments during the 45° cutting maneuver.

**Figure 3 sensors-24-03652-f003:**
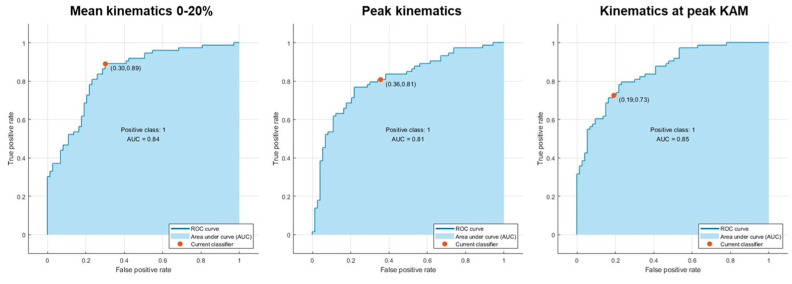
Area under the curve.

**Table 1 sensors-24-03652-t001:** Joint kinematics. Positive rotations: abduction, flexion, internal rotation.

Joint Kinematics (°)	Mean Kinematics 0–20%			Peak Kinematics			Kinematics at Peak KAM		
	High KAM	Low KAM	*p*-Value	High KAM	Low KAM	*p*-Value	High KAM	Low KAM	*p*-Value
Knee flexion	21.9 ± 10.5	20.2 ± 8.5	0.018	29.8 ± 11	27.9 ± 9.9	0.014	20.7 ± 11.4	20.1 ± 10.8	0.487
Knee abd–adduction	−1.6 ± 3.8	−1.5 ± 2.9	0.615	0.3 ± 4	0.2 ± 2.9	0.562	−1.9 ± 3.9	−1.5 ± 3.1	0.124
Knee internal–external rotation	−1.2 ± 5.9	−1.9 ± 5.3	0.122	2.1 ± 5.8	1.3 ± 5.1	0.052	−1.6 ± 6.2	−2.7 ± 5.7	0.012
Hip flexion	40.9 ± 9.9	37.5 ± 9.6	<0.001	43.1 ± 9.9	39.8 ± 9.7	<0.001	41.3 ± 10	38.2 ± 9.7	<0.001
Hip abd–adduction	10.4 ± 6.7	9.3 ± 6.6	0.023	11.6 ± 6.9	10.5 ± 6.9	0.029	10.7 ± 6.8	9.9 ± 6.8	0.106
Hip internal–external rotation	−1 ± 6.7	−0.8 ± 6.6	0.702	0.7 ± 6.7	1 ± 6.7	0.545	−0.8 ± 6.9	−0.4 ± 6.9	0.426
Ankle dorsi-plantarflexion	1.4 ± 10	−4.2 ± 12.7	<0.001	5.3 ± 8.4	0.9 ± 10.7	<0.001	0.9 ± 11	−4.8 ± 14.3	<0.001
Ankle inversion-eversion	−11.3 ± 6.5	−16 ± 8.9	<0.001	−8.2 ± 7.2	−12.4 ± 8.3	<0.001	−11.2 ± 7.2	−15.8 ± 9.4	<0.001
Ankle internal–external rotation	0.4 ± 8.6	3.8 ± 8.2	<0.001	3.0 ± 8.8	5.8 ± 8.2	<0.001	0.6 ± 8.7	3.7 ± 8.4	<0.001
Pelvis ante-retroversion	4.3 ± 4.4	4.9 ± 4.5	0.076	5.3 ± 4.4	5.9 ± 4.5	0.045	4.1 ± 4.5	4.5 ± 4.7	0.282
Pelvis tilt	5.5 ± 3.8	5.5 ± 3.5	0.828	5.9 ± 3.8	5.9 ± 3.5	0.836	5.4 ± 3.9	5.4 ± 3.5	0.841
Pelvis rotation	6.4 ± 3.3	6.7 ± 3.4	0.191	7.1 ± 3.4	7.6 ± 3.4	0.043	6.6 ± 3.4	7.2 ± 3.4	0.019

**Table 2 sensors-24-03652-t002:** Classification model.

	Mean Kinematics 0–20%	Peak Kinematics	Kinematics at Peak KAM
Model Validation			
N° features selected (PCA)	8/12	9/12	8/12
Best model	Fine Gaussian SVM	Fine Gaussian SVM	Fine Gaussian SVM
Accuracy (%)	78.8	74.7	77.9
AUC	0.85	0.81	0.84
Model Test			
Accuracy (%)	79.5	72.6	76.7
AUC	0.84	0.81	0.85
TPR (%)	89.0	88.8	72.6
PPV (%)	74.7	69.4	79.1

Note: AUC = area under curve; TPR = true positive rate; PPV = positive predictive values.

**Table 3 sensors-24-03652-t003:** Regression models.

	Mean Kinematics 0–20%	Peak Kinematics	Kinematics at Peak KAM
Model Validation			
N° features selected (PCA)	8/12	8/12	8/12
Best model	Gaussian Matern 5/2 GPR	Gaussian Matern 5/2 GPR	Gaussian Exponential GPR
RMSE (Nm/BW)	957	977	956
Model Test			
RMSE (Nm/BW)	937	1.078	936
R^2^	0.46	0.33	0.39

## Data Availability

The original contributions presented in the study are included in the article/Appendix A, further inquiries can be directed to the corresponding author.

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
