# Peer review of "Application of Machine Learning Methods to Investigate Joint Load in Agility on the Football Field: Creating the Model, Part I"

_sensors, 2024, doi:10.3390/s24113652_

Round 1

Reviewer 1 Report

Comments and Suggestions for Authors

The current study aimed to use machine learning (ML) methods with wearable sensors to predict knee joint loading during sport-specific agility tasks, focusing on knee abduction moments (KAM) in young female soccer players performing unanticipated sidestep cutting movements. The study found that classification models identifying players with high or low KAM were more effective than models predicting the actual peak KAM magnitude.

The study is of interest, but the introduction should cover more information, explaining clear structures, pathology, risks, traditional diagnosis, and other related points. Please find the suggestion below.

Introduction

1.      Please add the full term of ACL before using the abbreviation and add more details of this injury that are linked to the knee abduction moment, e.g., definition, pathology, traditional diagnosis, signs and symptoms, anatomy and ACL structures, risk factors, especially in females, movement-related injuries, etc.

2.      Please clarify the definitions and importance of high and low knee abduction moments (KAM).

3.      Please provide a reference to the contents, e.g., “On-field joint kinetics would allow for a more informative understanding of movement patterns in ecologically valid situations, as it describes the forces that cause the movement,” and explain more about the joint kinetics.

Materials and methods

1.      Although the full description of the Xsens MVN Analyze system was described in the previous study, the current study should briefly explain this device in the manuscript.

2.      Why were the 33rd and 67th percentiles of the peak KAM selected to be computed as boundaries of “low” and “high” KAM?

3.      Creating a figure to explain the data preparation and feature extraction would help the audience capture this process.

4.      Please add rationale or explain more about mean kinematics (0–20%), peak kinematics, and kinematics at peak KAM.

Discussion

1.      The study should list the limitations of the study, e.g., maturity, and others.

2.      Please add the discussion about clinical applications of the current findings.

Conclusion

The part should aware about the participants of the study that the results may not be possibly adopt to other conditions.

Author Response

Please see added Word document.

Reviewer 2 Report

Comments and Suggestions for Authors

In this paper, the authors explore machine learning (ML) methods for predicting knee joint loading during sport-specific agility tasks using data from kinematic analyses in a laboratory setting. They aim to refine ACL injury risk assessments by leveraging wearable sensors and advanced ML techniques. The study primarily focuses on identifying high and low knee abduction moments (KAM) through ML models, demonstrating a promising approach for on-field injury prevention and enhancing the understanding of joint mechanics outside traditional laboratory environments.

Revised Abstract:

The abstract should include the abbreviation 'ACL' (Anterior Cruciate Ligament) when first mentioned to clarify its meaning. This study addresses the limitations of laboratory-based biomechanical analyses in replicating real-world conditions. Despite the use of laboratory tests, the gold standard for on-field analysis involves wearable sensors and machine learning. However, the rationale behind comparing machine learning with peak performance analysis remains unclear. Additionally, the literature review appears insufficient with only seven citations; expanding this list could strengthen the justification by comparing various models and highlighting the unique contributions of this study.

Introduction:

The introduction should also define 'ACL' at its first mention. It discusses the limitations of traditional laboratory tests for ACL injury risks due to their failure to mimic real-world conditions. The justification for the research seems inconsistent as it criticizes the extrapolation of lab results to real-world scenarios yet relies on lab data. This study posits that wearable sensors combined with machine learning offer a more ecologically valid approach, although it does not clarify the novelty or necessity of comparing machine learning with traditional analysis methods when both use laboratory-derived data.

Methodology:

In the methodology section, it is mentioned that models were selected based on the best accuracy outcomes from five-fold cross-validation, but the paper only reports the best performance results of this process rather than the average across all folds, which could misrepresent the model's performance. Additionally, the balance of samples or classes and the justification for using cross-validation are not addressed. Mentioning important information in appendices without including them in the manuscript for review is also problematic. The paper should explain the use of cross-validation more thoroughly and ensure all relevant data is accessible within the main text or supplementary materials.

General Comments:

The paper's main weaknesses lie in its justification and methodological framing. The contradictory use of lab data, despite acknowledging its limitations, and the unclear added value of machine learning in this context, suggest a need for substantial revision. Expanding the literature review, clarifying the research's novelty, and enhancing the methodological transparency could significantly improve the manuscript's quality and contribution to the field.

Author Response

Please see attached Word document.

Round 2

Reviewer 1 Report

Comments and Suggestions for Authors

Thank you for addressing the concerns and suggestions provided in my review of your manuscript. I appreciate the effort you have put into revising the manuscript. The improvements you have made have significantly enhanced the clarity of the study.

Reviewer 2 Report

Comments and Suggestions for Authors

The authors assess all the comment and explain what was not clear in the first version.